# Enhancement of Anticancer Effects by Combining 5-Fluorouracil with Refametinib in Human Oral Squamous Cell Carcinoma Cell Line

Po-Chun Chen [1,2], Bor-Chyuan Su [3,4], Tien-Li Ma [5], Ying Chui Hong [6], Yu-Wen Chen [7], Thi Thuy Tien Vo [8], Luo-Yun Wu [9], Tzu-Yu Peng [6] ⦿, Ching-Shuen Wang [6] ⦿ and I-Ta Lee [6,*] ⦿

1    Translational Medicine Center, Shin-Kong Wu Ho-Su Memorial Hospital, Taipei 111045, Taiwan
2    School of Life Science, National Taiwan Normal University, Taipei 10663, Taiwan
3    Graduate Institute of Medical Sciences, College of Medicine, Taipei Medical University, Taipei 11031, Taiwan
4    Department of Anatomy and Cell Biology, School of Medicine, College of Medicine, Taipei Medical University, Taipei 11031, Taiwan
5    Department of Materials Science and Engineering, National Taiwan University, Taipei 106216, Taiwan
6    School of Dentistry, College of Oral Medicine, Taipei Medical University, Taipei 11031, Taiwan
7    School of Medical Laboratory Science and Biotechnology, College of Medical Science and Technology, Taipei Medical University, Taipei 11031, Taiwan
8    Faculty of Odonto-Stomatology, Hong Bang International University, Ho Chi Minh City 72500, Vietnam
9    School of Medicine, College of Medicine, Taipei Medical University, Taipei 11031, Taiwan
*    Correspondence: itlee0128@tmu.edu.tw; Tel.: +886-2-2736-1661 (ext. 5162); Fax: +886-2-2736-2295

**Abstract:** (1) Background: Oral squamous cell carcinoma (OSCC) is a significant health burden worldwide. This study aimed to determine the potentials of Refametinib, an orally bioavailable selective MEK1/2 inhibitor, to increase the effectiveness of 5-Fluorouracil (5-FU), a common cytotoxic drug, in the SCC4 cell line. (2) Methods: SCC4 cells were treated with increasing concentrations of 5-FU, either alone or in combination with Refametinib. The chemosensitivity to treatment was assessed via cell viability assay, microscopic observation, colony formation assay, and detection of apoptotic markers using Western blotting. The whole-cell expression and surface expression of programmed death-ligand 1 (PD-L1), an immune checkpoint protein which contributes to chemoresistance and affects treatment response, were also determined using Western blotting and flow cytometry, respectively. (3) Results: The combined treatment suppressed cell proliferation and promoted apoptosis in a more potent way than 5-FU treatment alone did. Additionally, MEK/ERK inhibition mitigated 5-FU-induced PD-L1 upregulation. (4) Conclusions: This is the first report of an enhanced anticancer effect and reduced PD-L1 expression for the combination of 5-FU with Refametinib in OSCC, suggesting a new promising combination strategy.

**Keywords:** 5-fluorouracil; apoptosis; extracellular signal-related kinase; MEK inhibitor; oral squamous cell carcinoma; programmed death-ligand 1

## 1. Introduction

Oral cancer, dominated up to 90% by oral squamous cell carcinoma (OSCC), is a highly prevalent cancer across the world. In general, OSCC has a poor prognosis because of its aggressive and metastatic nature, in which the recurrence arises in about 30% of cases, leading to a low 5-year survival and high mortality rates [1,2]. The use of anticancer drugs is imperative for the treatment of advanced cancers and metastases [3]. The arsenal of cancer pharmacologic treatment can be simply divided into three classes: cytotoxic chemotherapy for halting the rapid and uncontrolled proliferation of cancer cells, targeted therapy for directing the molecular targets involved in the tumorigenic processes, and immunotherapy for targeting the immune checkpoints implicated in the immune surveillance [4,5]. Among the panel of approved cytotoxic chemotherapeutic agents, 5-Fluorouracil (5-FU),

an antimetabolite drug, is commonly used to treat many cancers. The cytotoxicity of 5-FU is ascribed to the inhibition of essential biosynthetic processes and the incorporation into macromolecules (such as DNAs and RNAs) to suppress their normal function [6]. A past study demonstrated the anticancer effects of 5-FU in the cultured oral cancer cells, with apoptosis as the major mechanism that determined the therapeutic efficacy [7]. However, a phase III trial in patients with recurrent head and neck cancer reported that the overall response rate to 5-FU treatment was as low as 13% [8]. In addition to the low response rate, high drug resistance also presents a critical obstacle to the clinical use of 5-FU [9]. Therefore, identification of new therapeutic agents for their potential applicability combined with 5-FU in OSCC treatment may help reduce adverse effects and improve patient survival.

The mitogen-activated protein kinase/extracellular signal-regulated kinase (MAPK/ERK) pathway is crucial for cancer cell growth, survival, and differentiation [10]. By examining the biopsy specimens of OSCC, it was found that ERK1/2 was overexpressed and related to the proliferation in OSCC [11]. The extracellular signal-regulated kinase (MEK) is the downstream of BRAF in the RAS-RAF-MEK-ERK pathway, which is important for signal transduction to ERK [10]. The MEK inhibition provides an encouraging strategy for targeted therapy due to the high selectivity for its target pathway RAS-RAF-MEK-ERK, thereby halting cell proliferation and promoting apoptosis [12,13]. Furthermore, the combination of MEK inhibitors with other therapies, including chemotherapy, targeted therapy, and immunotherapy, is considered a hopeful prospect for clinical use [13]. Refametinib is a potent, highly selective, and allosteric inhibitor of MEK1/2. Its excellent pharmacokinetic profile and potency have led to its clinical development. [12,13]. In a subset of cancer patients, Refametinib has demonstrated lower toxicities compared to first-generation MEK inhibitors [14,15]. This suggests that it could be a promising new option for the treatment of ERK-associated malignancies. In several clinical trials, Refametinib has shown positive results as a monotherapy or in combination therapy, effectively inhibiting tumor growth in various solid tumors through MEK inhibition [16–18]. From this point of view, in the present study, we investigated the chemosensitivity of a human OSCC cell line (SCC4) to 5-FU, either alone or in combination with Refametinib, to determine whether this orally bioavailable selective MEK1/2 inhibitor may increase the effectiveness of 5-FU for OSCC treatment.

Improving our understanding of the mechanisms behind drug combinations would help us enhance the therapeutic efficacy of cancer treatments for patients. One crucial factor that allows cancer cells to multiply, spread, and metastasize is the absence of antitumor immunity [4]. The programmed death-ligand 1 (PD-L1) is a vital immune checkpoint that functions as an immunological brake in the modulation of T-cell activation, thereby compromising the immune surveillance. In cancer biology, PD-L1 has been recognized as crucial for cancer cells to evade the immune system [19,20]. There has been increasing evidence indicating the overexpression of PD-L1 in OSCC tumors as an unfavorable prognostic factor in OSCC [21,22]. Moreover, research reports have suggested that PD-L1 may play a crucial role in the development of chemoresistance [23,24], underscoring the significance of PD-L1 expression as an inadequate indicator of chemotherapy response. Multiple studies have explored the impact of 5-FU treatment on PD-L1 expression in diverse types of cancer cells [25–27]. Moreover, the MAPK pathway, particularly the MEK and the downstream effector ERK, is known to modulate PD-L1 expression in a variety of cancers [20]. Therefore, in this study, we further explored the effects of 5-FU, either alone or in combination with Refametinib, on the expression of PD-L1 in the SCC4 cell line. The present research may provide important insights for the design of combination regimens to improve treatment response and sustain treatment efficacy in OSCC, not only for their capacity of inhibiting cancer cell proliferation and inducing apoptosis, but also their propensity for modulating PD-L1 expression.

## 2. Materials and Methods

### 2.1. Reagents

The 5-Fluorouracil (5-FU), dimethyl sulfoxide (DMSO), phenazine methosulfate (PMS), and U0126 were purchased from Merck. Refametinib was obtained from MedChemExpress. The 5-FU, U0126, and Refametinib were dissolved in DMSO. The 3-(4,5-dimethylthiazol-2-yl)-5-(3-carboxymethoxyphenyl)-2-(4-sulfophenyl)-2H-tetrazolium, inner salt (MTS) was purchased from Promega Corporation.

### 2.2. Cell Culture and Cell Treatment

Human OSCC cells (SCC4) were maintained in DMEM/F12 medium (Thermo Fisher, Waltham, MA, USA), which was supplemented with 10% fetal bovine serum (FBS; Peak Serum, Wellington, CO, USA) and antibiotic-antimycotic (Thermo Fisher, Waltham, MA, USA), under a humidified atmosphere of 5% $CO_2$ at 37 °C. The cells were subjected to a medium change every 48 h and grown until 70–80% confluence was achieved for subsequent experiments. To evaluate the response of SCC4 cells to monotherapy or combination therapy, the cells were treated with varying concentrations of 5-FU (50, 100, 200, and 400 μM) for 24 h, with or without a 1-h pretreatment of Refametinib (50 nM).

### 2.3. Cell Viability Assay

The cell viability was assessed using the MTS/PMS assay as reported in our previous study [28]. In addition, the cell density was observed under a light microscope (Olympus, Tokyo, Japan).

### 2.4. Colony Formation Assay

To begin the experiment, cells were seeded at a density of 1000 cells per well in 6-well tissue culture plates and given 6 h to adhere. The cells were then treated with increasing concentrations of 5-FU (50, 100, 200, and 400 μM) for 24 h, with or without a 1-h pretreatment of Refametinib (50 nM). The medium was replaced the next day and subsequently every 2–3 days until day 8. At that point, the medium was removed, and the cells were rinsed with phosphate-buffered saline (PBS). The cells were fixed with 6.0% (*v/v*) glutaraldehyde and stained using 0.5% crystal violet. After washing the glutaraldehyde crystal violet mixture, the plates were scanned, and the number of colonies was quantified using ImageJ.

### 2.5. Western Blot

To extract the cell lysates, RIPA buffer (Merck, Rahway, NJ, USA) was used. The resulting lysates were separated via SDS-PAGE and then transferred onto PVDF membranes (Cytiva, Marlborough, MA, USA). The membranes were probed with a variety of antibodies purchased from Cell Signaling Technology, including cleaved PARP, PARP, caspase-3, pro-caspase-3, phospho-ERK1/2, ERK1/2, PD-L1, and β-actin.

### 2.6. Mitochondrial Membrane Potential Measurement

The mitochondrial membrane potential was determined using TMRE as described in our previous study [28]. The cells were treated with increasing concentrations of 5-FU (100, 200, and 400 μM) for 24 h with or without pretreatment with Refametinib (50 nM) for 1 h. Next, TMRE (100 nM) was added into the cells for 15 min. After rinsing the cells with PBS three times, TMRE intensities were determined using flow cytometry (Beckman Coulter, Brea, CA, USA).

### 2.7. Flow Cytometry

The cells were collected through trypsinization. After rinsing twice with PBS, the cells were incubated with PD-L1-FITC (Cell Signaling Technology, Danvers, MA, USA) for 1 h at 4 °C. Next, the cells were washed again with PBS three times. The levels of PD-L1 surface expression were assessed using flow cytometry (Beckman Coulter Life Sciences).

### 2.8. Statistical Analysis

Each experiment was independently conducted at least three times. The data were analyzed using GraphPad Prism software (GraphPad Prism, San Diego, CA, USA) and presented as mean ± standard deviation. Student's *t*-test was used to compare between two groups, while one-way ANOVA was utilized for multiple comparisons. Statistical significance was considered at $p < 0.05$.

## 3. Results

### 3.1. 5-FU Inhibits Cell Proliferation and Causes Apoptosis in SCC4 Cells

To evaluate the impact of 5-FU on the SCC4 cell line, various concentrations of 5-FU (50, 100, 200, and 400 μM) were administered for 24 h. The MTS/PMS assay indicated that the viability of SCC4 cells was significantly reduced through 5-FU treatment at all concentrations tested ($p < 0.05$) (Figure 1A). Microscopic observations were consistent with the results of the cell viability assay. As shown in Figure 1B, the control cells had a basic density level, while the 5-FU treatment reduced the number and size of cancer cell clusters. Increasing the concentration of 5-FU appeared to correspond with a progressive decrease in cell density in the SCC4 cells. Additionally, 5-FU treatment inhibited colony formation in a concentration-dependent manner (Figure 1C), indicating an anti-proliferative effect of 5-FU. To further elucidate the anti-cancer mechanism of action of 5-FU, the apoptotic markers were assessed. Western blot analysis revealed an increased expression of both cleaved PARP and caspase-3 in SCC4 cells following 5-FU treatment (Figure 1D). The relative densitometric graphs also demonstrated a significant dose-dependent increase in the cleavage of PARP and caspase-3 induced by 5-FU ($p < 0.05$) (Figure 1E). These findings therefore suggest that the major anticancer effects of 5-FU in SCC4 cells are anti-proliferation and pro-apoptosis.

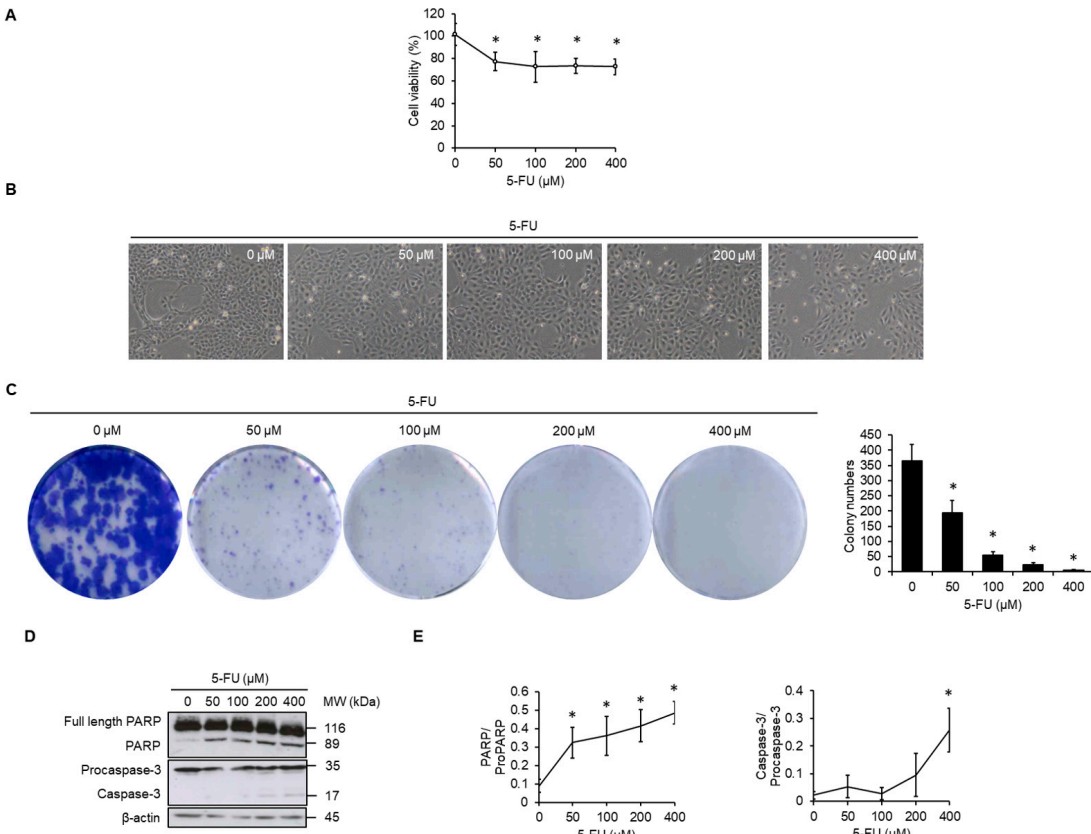

**Figure 1.** 5-FU inhibits cell proliferation and causes apoptosis in SCC4 cells. The cells were treated with increasing concentrations of 5-FU (50, 100, 200, and 400 μM) for 24 h. (**A**) The cell viability was

measured using MTS/PMS assay. (**B**) The cell density was observed under a light microscope (10×). (**C**) Representative images of colony formation assay and their quantitative analysis. The (**D**) Western blot analysis showing the protein expression of PARP, cleaved PARP, pro-caspase-3, and caspase-3. The (**E**) relative densitometric graphs showing PARP and caspase-3 cleavage. * $p < 0.05$, as compared to the control. All experiments were independently performed at least three times.

### 3.2. Refametinib Sensitizes SCC4 Cells to 5-FU Treatment for Suppressing Cell Proliferation and Promoting Apoptosis

Figure 2A,B demonstrates that treatment with the lowest tested concentration of 5-FU (50 μM) resulted in increased expression of phospho-ERK1/2 over time, while ERK1/2 expression remained unchanged in SCC4 cells, indicating activation of the oncogenic pathway through ERK1/2. To enhance the effectiveness of 5-FU treatment, we investigated the combination of 5-FU at various concentrations (50, 100, 200, and 400 μM) with Refametinib (50 nM), a selective MEK1/2 inhibitor that is orally bioavailable. Our results showed that the combination treatment improved the anticancer effects of 5-FU on SCC4 cells compared to the vehicle (5-FU plus DMSO). This was evidenced by a significant decrease in cell viability (Figure 2C), an increase in apoptotic markers (Figure 2D), a reduction in cell density (Figure 2E), and a decrease in colony formation (Figure 2F). Additionally, it is known that the mitochondrial membrane potential decreases during apoptosis [29]. As demonstrated in Figure 2G,H, the mitochondrial membrane potential in SCC4 cells was significantly lower for the combination treatment than for the vehicle. Overall, our findings suggest that the combination of 5-FU with Refametinib has a synergistic effect on inhibiting cell proliferation and promoting apoptosis in SCC4 cells.

### 3.3. Refametinib Mitigates 5-FU-Induced PD-L1 Expression in SCC4 Cells

Previous studies have suggested that cancer cells may use immune checkpoints as a survival mechanism to overcome chemotherapy-induced stress [30,31]. Furthermore, the MAPK/MEK/ERK pathway has been shown to regulate the expression of PD-L1 in various cancer types [20]. Given the observed activation of ERK1/2 by 5-FU in SCC4 cells, our study aimed to investigate the effect of 5-FU treatment on PD-L1 expression in SCC4 cells. As demonstrated in Figure 3A,B, PD-L1 protein levels were significantly increased in SCC4 cells treated with various concentrations of 5-FU (50, 100, 200, and 400 μM) for 24 h ($p < 0.05$). Additionally, our flow cytometry data indicated a significantly higher surface expression of PD-L1 in the treated cells compared to control cells ($p < 0.05$) (Figure 3C,D). To further explore the effect of combined treatment with 5-FU and MEK inhibition on PD-L1 expression in SCC4 cells, cells were pretreated with Refametinib (50 nM) prior to treatment with increasing concentrations of 5-FU (100, 200, and 400 μM). As shown in Figure 3E–G, PD-L1 expression in SCC4 cells treated with 5-FU was significantly reduced through the combined treatment with Refametinib, while the surface expression of PD-L1 remained nearly unchanged. To verify the pharmacologic specificity of these findings, a second MEK inhibitor, U0126, was used and produced similar results (Figure 3H–J), further supporting the conclusion that the MEK/ERK pathway modulates 5-FU-induced PD-L1 expression in SCC4 cells. Together, our results suggest that combined treatment with 5-FU and MEK inhibition could mitigate whole-cell PD-L1 expression in SCC4 cells.

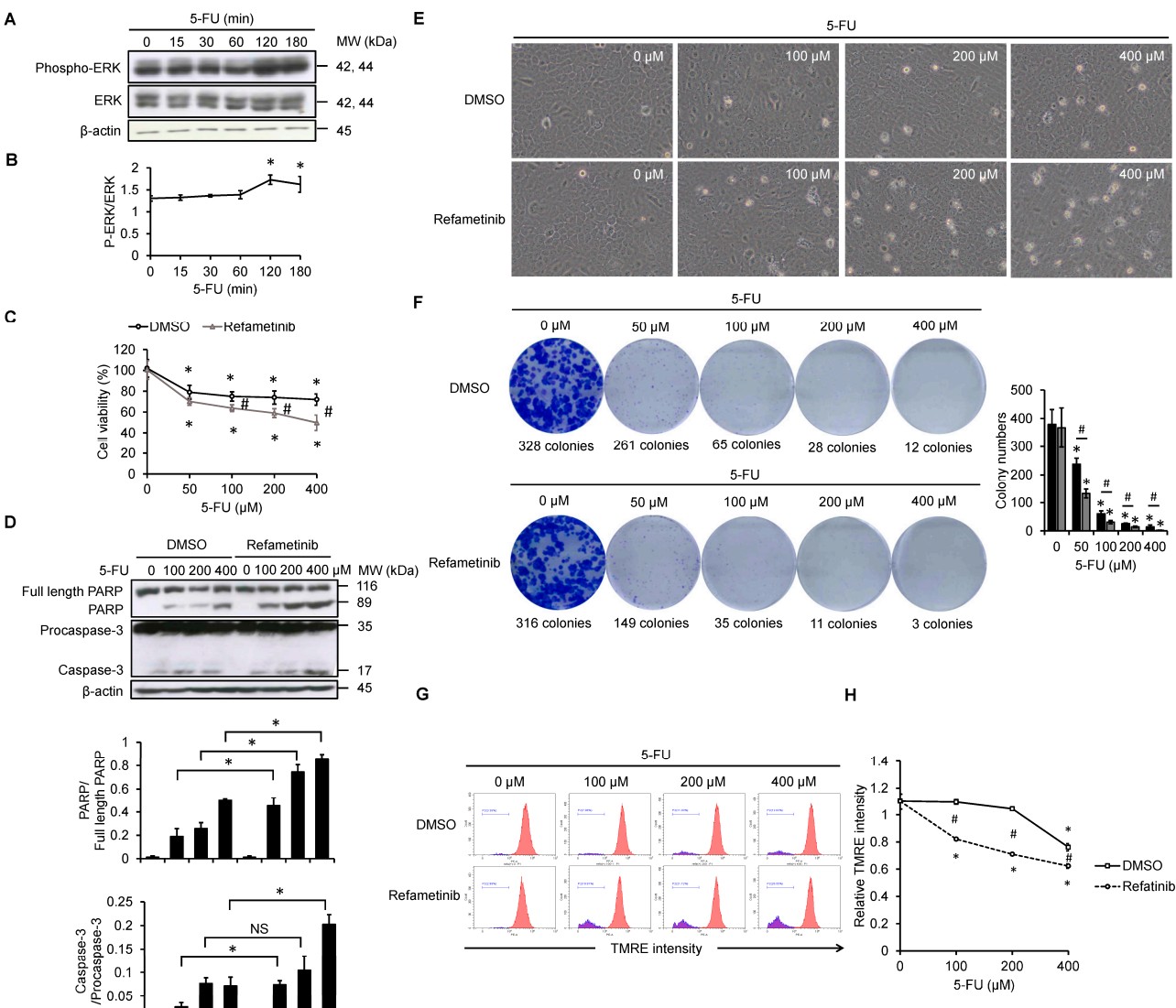

**Figure 2.** Refametinib sensitizes SCC4 cells to 5-FU treatment for suppressing cell proliferation and promoting apoptosis. (**A**,**B**) The cells were treated with 5-FU (50 μM) for the indicated time intervals. The (**A**) Western blot analysis and (**B**) relative densitometric graph showing the phosphorylation of ERK1/2. (**C**–**H**) The cells were pre-incubated with Refametinib (Re; 50 nM) for 1 h and then treated with increasing concentrations of 5-FU (50, 100, 200, and 400 μM) for 24 h. (**C**) The cell viability was measured using MTS/PMS assay. The (**D**) Western blot analysis showing the protein expression of PARP, cleaved PARP, pro-caspase-3, and caspase-3. The relative densitometric graphs showing PARP and caspase-3 cleavage. (**E**) The cell density was observed under a light microscope (10×). (**F**) Representative images of colony formation assay and their quantitative analysis. (**G**,**H**) Mitochondrial membrane potential was measured through flow cytometry using TMRE. * $p < 0.05$, as compared to the control. # $p < 0.05$, 5-FU alone vs. 5-FU + Refametinib. All experiments were independently performed at least three times.

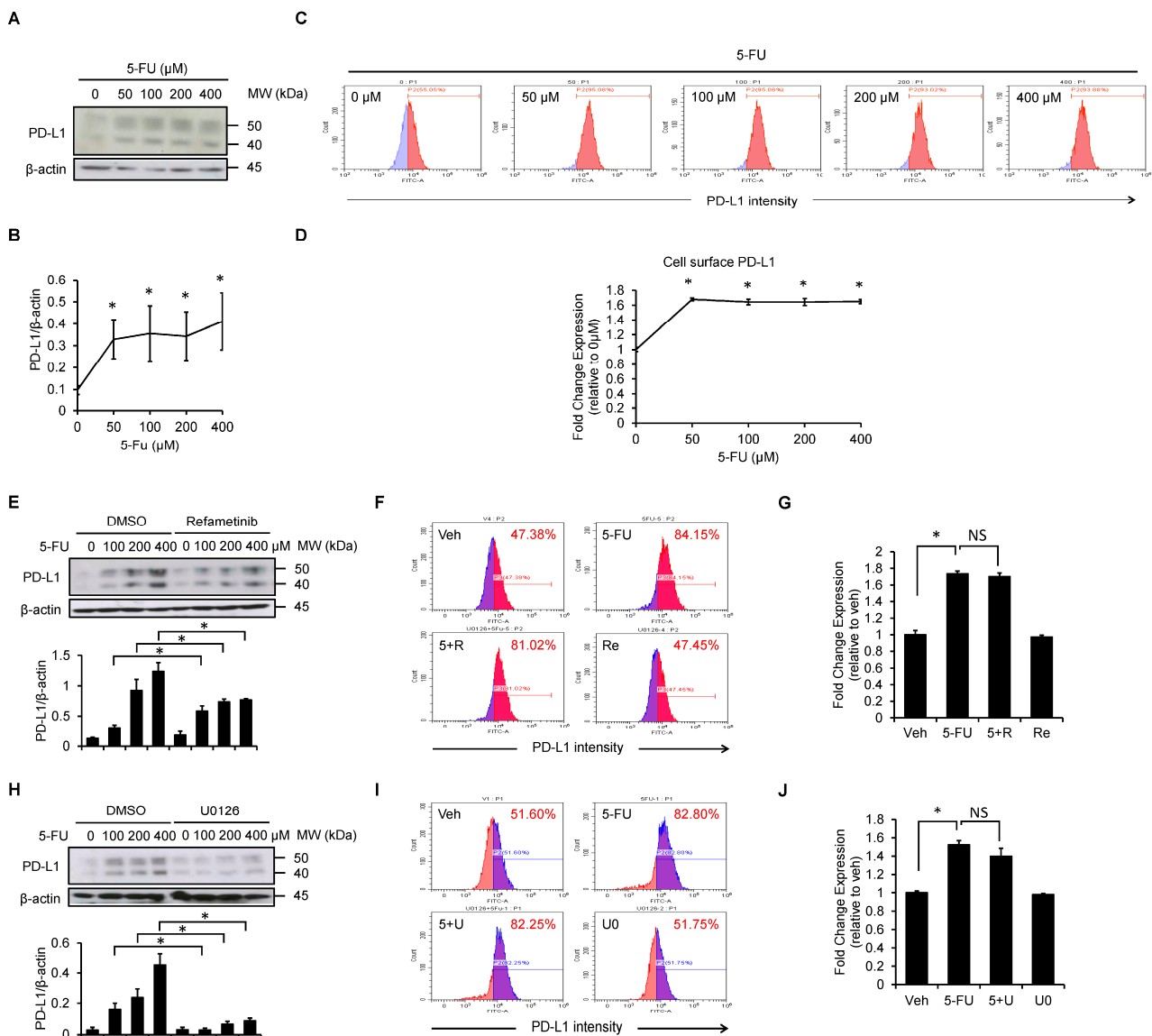

**Figure 3.** Refametinib mitigates 5-FU-induced PD-L1 expression in SCC4 cells. (**A–D**) The cells were treated with increasing concentrations of 5-FU (50, 100, 200, and 400 μM) for 24 h. The (**A**) Western blot analysis and (**B**) relative densitometric graph showing the total level of PD-L1 expression. The (**C**) flow cytometry graph and (**D**) relative fluorescence intensity showing the surface expression of PD-L1. (**E–J**) The cells were pre-incubated with (**E–G**) Refametinib (Re; 50 nM) or (**H–J**) U0126 (U0 or U; 50 nM) for 1 h and then treated with increasing concentrations of 5-FU (100, 200, and 400 μM) for 24 h. The (**E,H**) Western blot analyses showing the total level of PD-L1 expression. The (**F,I**) flow cytometry graph and (**G,J**) relative fluorescence intensity showing the surface expression of PD-L1. * $p < 0.05$, as compared to the control or as compared between the two indicated groups. NS: not significant; Veh: vehicle; 5 + R: 5-FU + Refametinib; 5 + U: 5-FU + U0126; Re: Refametinib; U0 or U: U0126. All experiments were independently performed at least three times.

## 4. Discussion

Chemotherapy is one of the most mainstreams methods used for treating OSCC, which is known to kill cancer cells mainly through apoptosis [1,32]. The burst of poly(APD-ribosyl)ation of nuclear proteins following PARP cleavage catalyzed by caspase-3 is required for apoptosis [33]. By using a human OSCC cell line (SCC4), this study suggested that 5-FU, a cytotoxic chemotherapeutic agent, can induce OSCC cell death through apoptosis followed by caspase-3-mediated PARP cleavage, which is consistent with a previous research

reporting the caspase-dependent chemocytotoxicity of 5-FU in the cultured oral cancer cells [7]. Our results also demonstrated the anti-proliferative effect of 5-FU treatment in SCC4 cell line. Furthermore, the dose-response experiments indicated that a higher drug concentration was correlated with lower cell viability, less colony formation, and higher levels of apoptotic markers. However, it is important to notice a therapeutic tightrope regarding the balance between the maximum cancer cell death with the tolerable toxicity. Therefore, the combination of anticancer drugs may help to achieve a better therapeutic efficacy than the use of a single agent without exceeding the maximum tolerable dose of each. Moreover, drug combinations may provide some benefits to prevent the development of drug resistance [5,34].

The MAPK pathway comprises different signaling cascades that transduce diverse extracellular signals to the nucleus upon the phosphorylation activation, thereby regulating the expression of a vast array of molecules involved in cell survival, proliferation, apoptosis, invasion, metastasis, and many cellular processes. ERK1/2, a critical component of MAPK pathway, plays a crucial role in promoting cell survival and proliferation [10]. In this research, ERK1/2 phosphorylation was significantly enhanced through 5-FU treatment in SCC4 cell line. A previous study reported the overexpression of ERK1/2 in the biopsy specimens of OSCC, which was associated with the proliferation in OSCC [11]. Thus, ERK1/2 activation through 5-FU may promote cancer cell proliferation, indicating the paradoxically pro-survival effect of 5-FU that counteracts its principal cytotoxic activity and compromises its efficacy. Logically, pharmacologic blockage of the pathway with MEK inhibitors may inhibit 5-FU-induced ERK activation and downstream effects. To elucidate this assumption, a combination of 5-FU with an orally bioavailable selective MEK1/2 inhibitor, Refametinib, was used to treat the SCC4 cell line. Our results found that Refametinib could sensitize SCC4 cells to 5-FU treatment for synergistically inhibiting cell proliferation and promoting apoptosis. Similarly, preclinical evidences have demonstrated that MEK/ERK pathway inhibition increased the sensitivity to 5-FU in different colorectal cancer cell lines and murine colorectal tumor xenograft models [35–37]. Therefore, a combination of 5-FU with MEK/ERK inhibition may be a prospective approach in order to enhance the effectiveness of cancer pharmacologic therapies for OSCC treatment.

The transmembrane protein PD-L1, a co-inhibitory factor of the immune response, is considered pro-tumorigenic because its expression in cancer cells can render these cells to escape the antitumor immunity [19]. There has been increasing evidence to indicate PD-L1 expression in tumors as a poor prognostic marker for OSCC patients [21,22]. It was reported that various cytotoxic drugs, including paclitaxel, etoposide, and 5-FU, induced PD-L1 surface expression in human breast cancer cells to promote PD-L1-mediated T-cell apoptosis, suggesting a relationship between chemotherapy and cancer immunoresistance [27]. Another study concluded that 5-FU enhanced PD-L1 expression in head and neck squamous cell carcinoma cells [25]. Similarly, our research also provided the evidence linking PD-L1 upregulation, both the whole-cell expression and surface expression, with 5-FU treatment in SCC4 cell line, indicative of a positive association. The literature has indicated the MAPK/ERK signaling pathway as a crucial mechanism that regulates the expression of PD-L1 in cancer cells [19,20], contributing to the chemoresistance [38]. Therefore, the inhibition of the MAPK/ERK signaling pathway may have a double effect on cancer cells, reducing PD-L1 expression and suppressing cancer cell survival. A previous study demonstrated the increases in ERK phosphorylation and PD-L1 expression in esophageal squamous cell carcinoma cells following standard chemotherapy treatments, which were attenuated by the inhibition of MEK [39]. Similarly, our research reported that treatment with 5-FU combined with the blockage of MEK/ERK signaling pathway reduced PD-L1 expression in the SCC4 cell line because the inhibition of MEK1/2 using either Refametinib or U0126 reduced the high levels of PD-L1 expression in the 5-FU-treated SCC4 cells. In contrast with the substantial reduction in the total levels of PD-L1 protein, the significant expression of PD-L1 on the surface of the 5-FU-treated cells remained nearly unchanged following the inhibition of MEK1/2, as determined with flow cytometry. In fact,

the literature has reported that the profile expression of PD-L1 includes membrane-bound, extracellular soluble, and exosomal forms [40]. Thus, the differences between two analyses herein may reflect the presence of non-surface forms of PD-L1 and the influences of dead or dying cells upon treatment which could be excluded from the flow analysis. Moreover, PD-L1 expression is regulated by multiple factors at different levels of transcription, post-transcription, and post-translation [20]. Therefore, our findings still support a rationale for the combination of 5-FU with MEK/ERK-targeted therapy to boost the anticancer effects in OSCC by mitigating PD-L1 upregulation, which represents a key mechanism for the immune evasion and chemoresistance.

### 5. Conclusions

PD-L1 expression is an important factor that determines the cancer treatment response. The present study indicates that treatment with 5-FU, a cytotoxic chemotherapeutic drug of choice for many cancers, can promote PD-L1 expression in a cell model of OSCC. Our research also suggests that ERK1/2 signaling pathway may in part contribute to PD-L1 upregulation during 5-FU treatment. Logically, the next step to progress OSCC treatment may therefore involve the combination of this cytotoxic drug with MEK/ERK-targeted therapy such as the orally bioavailable selective MEK1/2 inhibitor Refametinib. However, the challenge is to determine the balance for taking full advantage of the combination in a specific context of disease. More studies are required to determine the drug choice, combination, and sequence, as well as to understand the underlying mechanisms.

**Author Contributions:** Conceptualization, P.-C.C., Y.C.H. and I.-T.L.; Methodology, P.-C.C., B.-C.S., T.-L.M., Y.C.H., Y.-W.C. and T.-Y.P.; Software, B.-C.S., T.-L.M., Y.C.H., Y.-W.C. and L.-Y.W.; Validation, T.-L.M.; Formal analysis, P.-C.C., B.-C.S., Y.C.H., Y.-W.C., L.-Y.W. and T.-Y.P.; Investigation, P.-C.C., B.-C.S., T.-L.M., Y.C.H. and I.-T.L.; Resources, P.-C.C. and I.-T.L.; Data curation, P.-C.C., B.-C.S., Y.C.H., Y.-W.C. and L.-Y.W.; Writing—original draft, P.-C.C., T.T.T.V. and I.-T.L.; Writing—review & editing, T.T.T.V. and I.-T.L.; Visualization, B.-C.S., T.-L.M., T.-Y.P., C.-S.W. and I.-T.L.; Supervision, P.-C.C., B.-C.S., T.T.T.V. and I.-T.L.; Project administration, P.-C.C., T.-L.M., T.T.T.V., C.-S.W. and I.-T.L.; Funding acquisition, P.-C.C., B.-C.S. and I.-T.L. All authors have read and agreed to the published version of the manuscript.

**Funding:** This work was supported by the Shin-Kong Wu Ho-Su Memorial Hospital and Taipei Medical University, grant number SKH-TMU-110-03 and the college of Oral Medicine, Taipei Medical University, grant number TMUCOM2022B02.

**Institutional Review Board Statement:** Not applicable.

**Informed Consent Statement:** Not applicable.

**Data Availability Statement:** Data is contained within the article.

**Acknowledgments:** We thank You-Syun Jheng for her technical assistance.

**Conflicts of Interest:** The authors declare no conflict of interest.

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
