# Peer review of "Enhancement of Anticancer Effects by Combining 5-Fluorouracil with Refametinib in Human Oral Squamous Cell Carcinoma Cell Line"

_applsci, doi:10.3390/app13074340_

Round 1

Reviewer 1 Report (Previous Reviewer 3)

The Paper has addressed most of the problems raised from previous time. The presentation is clearer than it used to be. I think the paper could be admitted.

Author Response

Thank you very much for your review.
We will continue to work hard on research.

Reviewer 2 Report (Previous Reviewer 1)

Dear authors,

Thank you for your kind response to my previous review.

The authors have responded adequately to most reviewer suggestions.

The manuscript has been much improved and is in a nice condition now. I think this manuscript is acceptable.

Good luck with further research.

Author Response

Thank you very much for your review.
We will continue to work hard on research.

Reviewer 3 Report (New Reviewer)

The manuscript entitled "Enhancement of anticancer effects by combining 5-fluorouracil with Refametinib in human oral squamous cell carcinoma cells" aims to demonstrate that Refametinib increases the anticancer effect of 5-FU when both compounds are associated. 

The authors proved first that 5-FU inhibits cell proliferation and causes apoptosis in SCC4 cells;

Then, they showed that Refametinib sensitizes SCC4 cells to 5-FU treatment for suppressing cell proliferation and promoting apoptosis;

Finally, they evidenced that Refametinib mitigates 5-FU-induced PD-L1 expression in SCC4 cells.

The study is well-designed and suitably conducted to the final.

The introduction is comprehensive, providing an appropriate background. The methods are well described, with suitable references.

The Results section has relevant figures.

The Discussion section extensively analyzes the studied mechanisms and their importance in the anticancer effect.

The following comments are available below:

1. In the title, the human oral squamous cell carcinoma cell line may be better. Please, check and correct.

2. For better visibility and understanding, the reviewer suggests the augmentation of microscopic and flow-cytometry images such as 1B and 1C, 2E, 2F, and 2G, 3E, 3F, and 3I.  

3. Moreover, the authors are encouraged to present the results in more detail, thus giving more significant value to their work.

Author Response

Thank you very much for your review.
1. According to Reviewer’s suggestion, we have modified the title.

2. According to Reviewer’s suggestion, we have done our best to increase the resolution of these images. We hope you can accept the review.

3. According to Reviewer’s suggestion, we have done our best to rewrite our "Experimental Results" section. 

This manuscript is a resubmission of an earlier submission. The following is a list of the peer review reports and author responses from that submission.

Round 1

Reviewer 1 Report

Dear authors,

The article: "Enhancement of anticancer effects by combining 5-fluorouracil with Refametinib in human oral squamous cell carcinoma cells" is a paper showing that the combination of 5-Fu and Refametinib enhanced 5-Fu's efficacy as a cytotoxic anticancer agent and suppressed its paradoxical side effect of increasing PD-L1 expression.

If it is possible to enhance the therapeutic effect of 5-Fu not only by increasing its efficacy as a cytotoxic anticancer agent, but also by suppressing PD-L1 expression, this is an important issue for overcoming resistance to immunotherapy and improving the prognosis of oral cancer patients, which makes this study very interesting as a topic.

However, the study needs to be improved prior to being considered for publication.

Major

・Although only one cell line was used in this study, two or more cell lines should be studied to increase the reliability of the study.

・Fig. 1A does not seem to show any results that can be described as dose-dependent. Please provide clear evidence to show dose-dependent changes.

・It is difficult to identify morphological changes in Fig. 1B and Fig. 2E. We recommend using another method to show morphological changes, such as fluorescence immunostaining, to show clear changes. It is then recommended to show a more magnified image.

・You state that proliferative capacity was inhibited from the results of colony formation assays, but colony formation assays in cancer cells are commonly considered to test for anchorage-independence and sensitivity to chemotherapy. Please explain in detail how the colony formation assay, which shows cell proliferative capacity, was performed, including how the number of colonies was measured. Also, manually counting the number of colonies in a 6-well plate is difficult and impractical in terms of experimental reproducibility. It is recommended that other methods of quantification are used to show significant differences, accompanied by graphs.

・You describe a synergistic effect of the combination of 5-FU and Refametinib, but the results do not seem to show any evidence of synergy. Is it possible that the synergistic effect may be due to inhibition of PD-L1 expression rather than enhanced cytotoxic effect? As this is the subject of this study, it is strongly recommended to avoid overestimation and provide your experimental results as evidence.

Mainor

・There is a mention in line 152 regarding an increase in floating cells, but no evidence is given for the results.

・Please provide results on cytotoxicity and ERK suppression when Refametinib is used alone.

・Please quantify Western blot results in Fig.2D,3E,3H.

Reviewer 2 Report

This is a fine piece of research.  Combination chemotherapy is becoming the method of choice.  Using 5-fu with Refametinib against OSCC is an excellent approach which might lead others to try such a method.  The lack of antitumor immunity is a critical I have read recently of a few such experiments and this work adds to the list.

Reviewer 3 Report

Overall, this paper does not show solid evidence for synergistic effect of 5-FU and Refametinib. The result could be due to short treatment time and unsuitable dosage. Western blot and flow cytometry results should be repeated, or to use other more quantitative methods for a more accurate result. The research report was poorly formatted with inadequate language efforts.

Specific Problems:

All 5-Fu should be 5-FU

Fig 1A treatment for 24h is not long enough, thus the result is not significant (no dose response when your dose changed from 50 to 400).

Fig. 2A: ERK band is not significantly changed, and the conclusion is in doubt.

Synergy experiment should be done with a matrix of doses (A&B) and analyze the data with https://synergyfinder.fimm.fi/

Fig1c and 2f : should add a graph to show the dose vs colony number

Fig 3G&J: @ should be replaced by “NS.” (Usually, in other papers)

Fig3: I wonder the logic of measuring PD-L1 level for your paper. PD-L1 is associated with cancer immune escape, but in this paper, PD-L1 level is upregulated when you treat cells with 5-FU. Does that mean 5-FU treatment will help cancer cell escape T cell immune response? And adding MEK inhibitor would down regulate PD-L1 to boost anti-tumor immune response? This point should be mentioned.

References: 

Ref 39. Incomplete author names. 

Ref 12,16,17,18,26,30,26,28,40 Incomplete for the author list.